# MoGeo, a Mobile Application to Promote Geotourism in Molise Region (Southern Italy)

**Francesca Filocamo**, **Gianluigi Di Paola** *, **Lino Mastrobuono** and **Carmen M. Rosskopf**

Department of Biosciences and Territory, University of Molise, Contrada Fonte Lappone, 86090 Pesche (IS), Italy; francesca.filocamo@gmail.com (F.F.); lino.mastrobuono@ordinegeologimolise.it (L.M.); rosskopf@unimol.it (C.M.R.)
* Correspondence: gianluigi.dipaola@unimol.it; Tel.: +39-0874-404168

**Abstract:** Geotourism represents a powerful and new form of sustainable tourism that has rapidly expanded worldwide over the last decades. To promote it, the use of digital and geomatic tools is becoming of increasing importance. Especially mobile information represents one of the most efficient and smart ways to bring geotourism closer to a wide audience. This applies in particular to rural and inner areas, where the exploitation of geoheritage can represent a crucial resource for eco-friendly and sustainable tourism development. With the aim to promote geotourism on a regional scale, we have implemented a mobile devise application for Molise region, tested in the Alto Molise area. This application, called MoGeo App, aims at providing diversified geotourism information that combines geologic attractions (geosites and geologic itineraries) with other possible tourist attractions (other sites of natural and cultural interest), to respond to differentiated interests and needs of a wide audience. Besides geotourism purposes, the structure of MoGeo App can be used also for other purposes such as educational targets, by adapting contents and language. It appears to be a flexible, easily updatable digital tool, adaptable to various target groups, as well as other regional contexts, both inside and outside of Italy.

**Keywords:** geology-based tourism; geosites; geoheritage; cultural heritage; web-GIS; smartphone; Alto Molise

## 1. Introduction

Geotourism, understood as a form of tourism that specifically focuses on geology and landscape [1–6], represents a powerful and relatively new form of sustainable tourism [7,8]. It has rapidly expanded over the last decades [4–9] all around the world [5,7,10] and become a substantial part of the overall tourism offer [4], as well as an important research direction (e.g., [6,10]).

Geotourism focuses on geoheritage and therefore on geosites that are the most essential part of it [5,9,11–13]. It represents an important alternative or integration to more traditional forms of tourism, such as sun and sand tourism, and cultural tourism. Furthermore, it can become an important economic resource for countries and regions that are characterized by a rich natural heritage and great geodiversity (e.g., [8,14–18]). Especially in rural and inner areas, the exploitation of geoheritage can represent a crucial resource for eco-friendly and sustainable tourism development. Here, traditional and mass tourist destinations are generally scarce or lacking, and major tourist attractions are typically related to the geodiversity and naturalness of the landscape [19–23]. It is in such areas that the geo-landscape and the related geodiversity, biodiversity and cultural values [5,24,25] become important drivers for the local and regional economy.

Geotourism, both in a pure sense and characterized by the integrated fruition of geological sites and other places of interest (natural, historical, archaeological, etc.) [20,26,27], can be of high interest

for various target groups [23] and especially for families. Families, in fact, must often turn to tourist forms and offers that take into account differentiated interests and needs of family members, including simple enjoyment and relaxation, as well as the desire to receive stimuli to learn something about and better appreciate the natural and cultural landscape [28–30].

The promotion, organization and exploitation of tourist offers now make increasing use of digital information sources (especially tourism websites) [31–33], not only in relation to the choice of tourist offer, travel and accommodation organisation, etc., but also in relation to the description and illustration of tourist attractions. This is particularly true also for geotourism [33]. Especially in the field of geoheritage conservation, management and promotion, the use of internet, digital and geomatic tools is becoming of increasing importance [33–37], even if geotourism is still being promoted little online [33]. Among such digital tools, especially the use of mobile phone devises and related applications to receive tourist information has experienced very recently a wide and rapid diffusion [38–41]. However, despite of the elevated potential and specific strengths of digital mobile tools (easy to transport, multi-sensorial, etc.) [42], there is still little use in the field of geotourism promotion and tour guiding [33,42–45]. Especially the scarce diffusion of app-based mobile tour guides (AMTG) [46] is surely at least partially due to the possible limitations related to the use of mobile phone applications [44], which must be carefully considered and best reduced. While keeping in mind such limitations, it is clear that mobile phones are already or will quickly be the most important interface between visitors/tourists and tourism contexts.

In geotorism field [34], three main types of applications can be distinguished: (1) applications that are based on georeferencing and mapping of geotourism assets, taking advantage in particular of recent developments in web mapping and mobile data access of maps (e.g., [42]); (2) applications that return 3D models based on photogrammetry, laser scanning or real-time observations of natural phenomena through a webcam (e.g., [47]); (3) applications that make interpretations using Augmented Reality (AR), a process that enriches discovery through digital media or virtual reality technologies creating a virtual universe that helps to imagine everything (e.g., [48]). These typologies can also be combined among them and coexist together [34].

Convinced that mobile apps can strongly support the promotion of geotourism, especially in rural and inner, less urbanized areas, we have implemented a mobile phone application that is illustrated in this paper. This application refers to the first type and aims at providing diversified "geotourism" information that includes not only the geologic attractions (geosites and geologic itineraries) but also other possible tourist attractions (other sites of interest) to respond in this way to different interests and needs of users, especially of families.

The application we propose should operate on a regional scale or on smaller areas. To develop it, we have chosen the Molise region. This region offers on a relative small area a representative view of the major geological-geomorphological and landscape features that typically characterize the central-southern Apennines [18]. It is characterized by a high geodiversity, and a regional inventory of geosites is already available [49]. However, the divulgation of geosites in the Molise region is only scarcely developed and restricted to some notions about geosites in leaflets directly distributed by the regional "Service for Tourist Promotion and Relations with Molisans in the World", and the online information provided by some institutional websites [50–52]. Regarding the promotion of geotourism at the regional or sub-regional scale, specific products are totally lacking. Some associations, essentially the ones promoting respectively the rocky spurs, so-called Morge, in Molise region, the Guardiaregia-Campochiaro Oasis and the Collemeluccio-Montedimezzo Alto Molise Biosphere Reserve [53–55], provide only short information about some geosites and only in Italian, and/or promote trekking activities and itineraries that include some of them. Besides this, some scientific and popular publications (e.g., [18,20,21,56,57]) have been produced with the aim of promoting geosites and geotouristic itineraries of specific areas of the Molise region, such as the Matese area, the Mt. Mainarde-Alto Volturno area and the Alto Molise area.

To start our project, we have selected the Alto Molise area (AM, Figure 1a), one of the seven major physiographic units in which the Molise Region has been subdivided mainly based on geological and orographic-hydrographic characteristics [21,49,58]. The main reasons for this choice are: the Alto Molise area has a small surface area, but is characterized by (i) a geological-geomorphological context that is representative for the history of the central-southern Apennines [18], and (ii) a high geodiversity and a significant number of geosites of different scientific interest. The Alto Molise area is also rich in natural protected areas, as well as important architectural, historical and archaeological sites, and retains important traces of ancient agro-pastoral traditions and crafts. In fact, the data collected during the regional geosite inventory activities [49] and several studies carried out to promote its geotouristic exploitation [18,20,57] have allowed to point out also its geographical features and traditions, as well as historical, archaeological and faunistic-floristic aspects, highlighting its rich natural and cultural heritage [57]. Therefore, the knowledge already available makes the Alto Molise area a good starting point to develop the application.

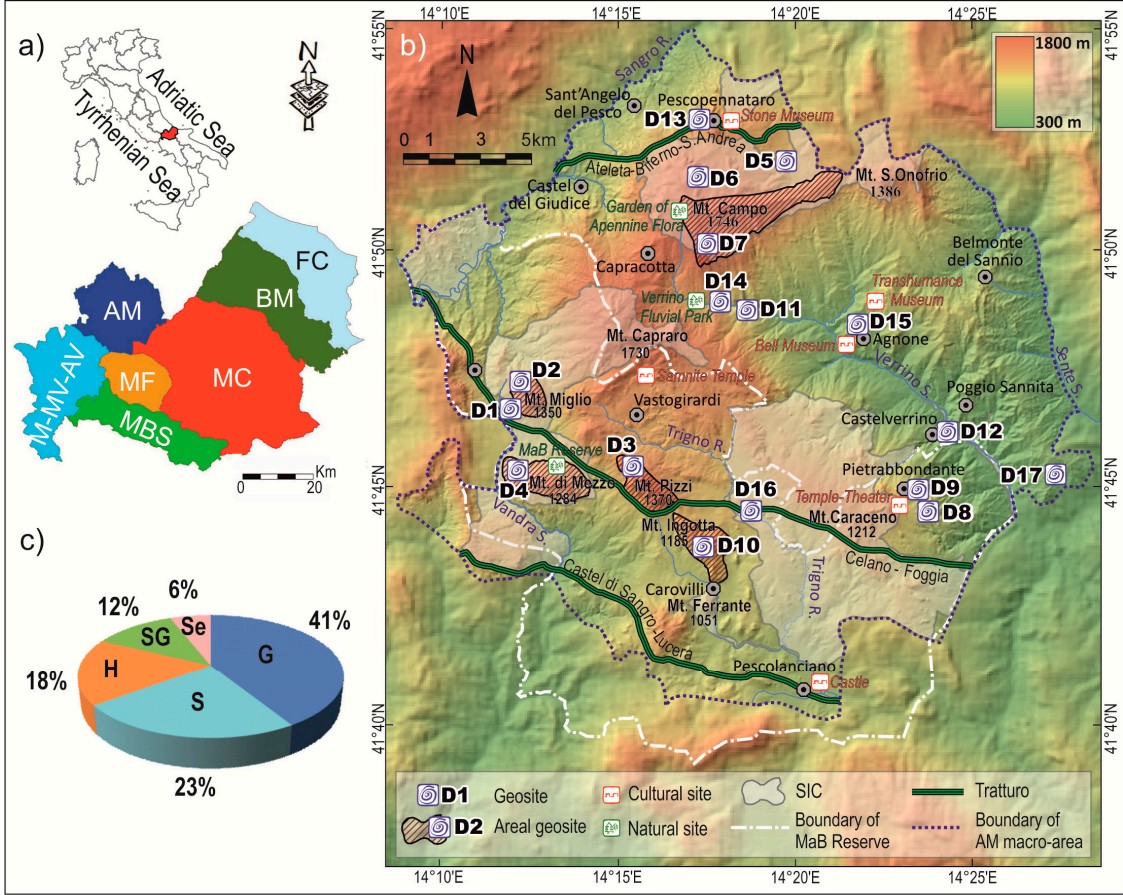

**Figure 1.** (**a**) The Molise region and its subdivision in seven macro-areas. M-MV-AV = Mainarde-Monti di Venafro-Alto Volturno, MBS = Matese-Conca di Boiano-Sepino, MF = Montagnola di Frosolone, AM = Alto Molise, MC = Molise Centrale, BM = Basso Molise, FC = Fascia Costiera; (**b**) Altitude and drainage network map of the Alto Molise macro-area with the location of the 17 assessed geosites and the other sites of interest; (**c**) Pie chart illustrating the assessed types of geosites and related percentages, based on primary scientific interests (G = Geomorphology, S = Stratigraphy, SG = Structural Geology, Se = Sedimentology, H = Hydrogeology).

## 2. The Test Area

The Alto Molise area (AM, Figure 1a) has a surface area of 452 km$^2$ and is located in the northwestern sector of the Molise region [20,21]. It is a macro-area particularly rich in geosites,

in total 17 (Figure 1b, Table 1) of the 100 assessed through the regional geosite inventory carried out by the Department of Biosciences and Territory of the University of Molise in partnership and on behalf of the Molise Region [18,49]. During this inventory project, the Molise geosites have been assessed by using a quantitative method that allowed to determine their representativeness, rarity, the scenic-aesthetic, historical-archeological-cultural, and vulnerability values. Based on the assessed values, the used method allowed to calculate for each geosite the so-called "Intrinsic Value of the Site of Geological Interest" (IVSGI) [18], corresponding to the weighted sum of representativeness, rarity, and scenic-aesthetic values.

**Table 1.** Code, name, main scientific interests, and estimated relevance of the Alto Molise geosites.

| ID | Name | Scientific Interests | Relevance |
|---|---|---|---|
| D1 | Capo di Vandra spring | Hydrogeology | Regional |
| D2 | Mt. Miglio monocline | Geomorphology | Regional |
| D3 | Mt. Pizzi deep-seated gravitational slope deformation | Geomorphology | Regional |
| D4 | Mt. di Mezzo morphostructure | Geomorphology | Regional |
| D5 | Rio Verde springs | Hydrogeology | Regional |
| D6 | Rocky ledge of the S. Luca Hermitage | Geomorphology | Regional |
| D7 | Mt. Campo –Mt. S. Nicola monocline | Geomorphology Structural geology | National |
| D8 | I Colli Agnone Flysch outcrop | Stratigraphy Sedimentology | Regional |
| D9 | Morge of Pietrabbondante | Stratigraphy Geomorphology | Regional |
| D10 | Mt. Ingotta anticline | Structural geology Geomorphology Paleontology | Regional |
| D11 | Verrino Stream spring | Hydrogeology | Local |
| D12 | Agnone Flysch outcrop of Castelverrino village | Stratigraphy Sedimentology | Regional |
| D13 | Pescopennataro fault planes | Structural geology | National |
| D14 | Verrino Stream waterfalls | Geomorphology | Regional |
| D15 | Cogoli walls of Agnone village | Stratigraphy Sedimentology | Regional |
| D16 | Vomero Resurgence | Geomorphology | Regional |
| D17 | Limestones with pyrite nodules of Poggio Sannita | Sedimentology | Regional |

The Alto Molise geosites, based on their primary scientific interest, refer mainly to the geomorphology or stratigraphy type (Figure 1c), although many of them have multiple scientific interests (for example Geomorphology-Structural geology, Table 1). The estimated relevance of the Alto Molise geosites [57] is mostly regional, national or local, instead, for three of them (Table 1). Most of the Alto Molise geosites (14) are already included in the Italian Geosites Inventory managed by ISPRA [52].

The Alto Molise area is characterized by a mainly mountainous and hilly landscape, whose major peaks are Mt. Campo (1746 m) and Mt. Capraro (1730 m) (Figure 1b), and by a few low-lying flat areas, such as the valley floors of the Trigno and Sangro rivers, which are the major watercourses in this area (Figure 2). Most of the outcropping bedrock is part of a sedimentary succession referred to the Molise pelagic basin domain, Oligocene to Late Miocene in age, which is interposed between the Apennine [59] and the Apulia carbonate platforms [60,61]. This succession is mainly composed of four geolithological units (see Figure 2): Varicolored clays and marls (10), Limestones and marls (9), Clayey marls and limestones (7) and Siliciclastic deposits (6). Units 7 and 9 form the main mountain ridges, and Unit 10 the hilly areas surrounding the mountain ridges. Finally, Unit 6 crops out widespread in the valley incisions of the Sangro River and Verrino Stream (Figure 2).

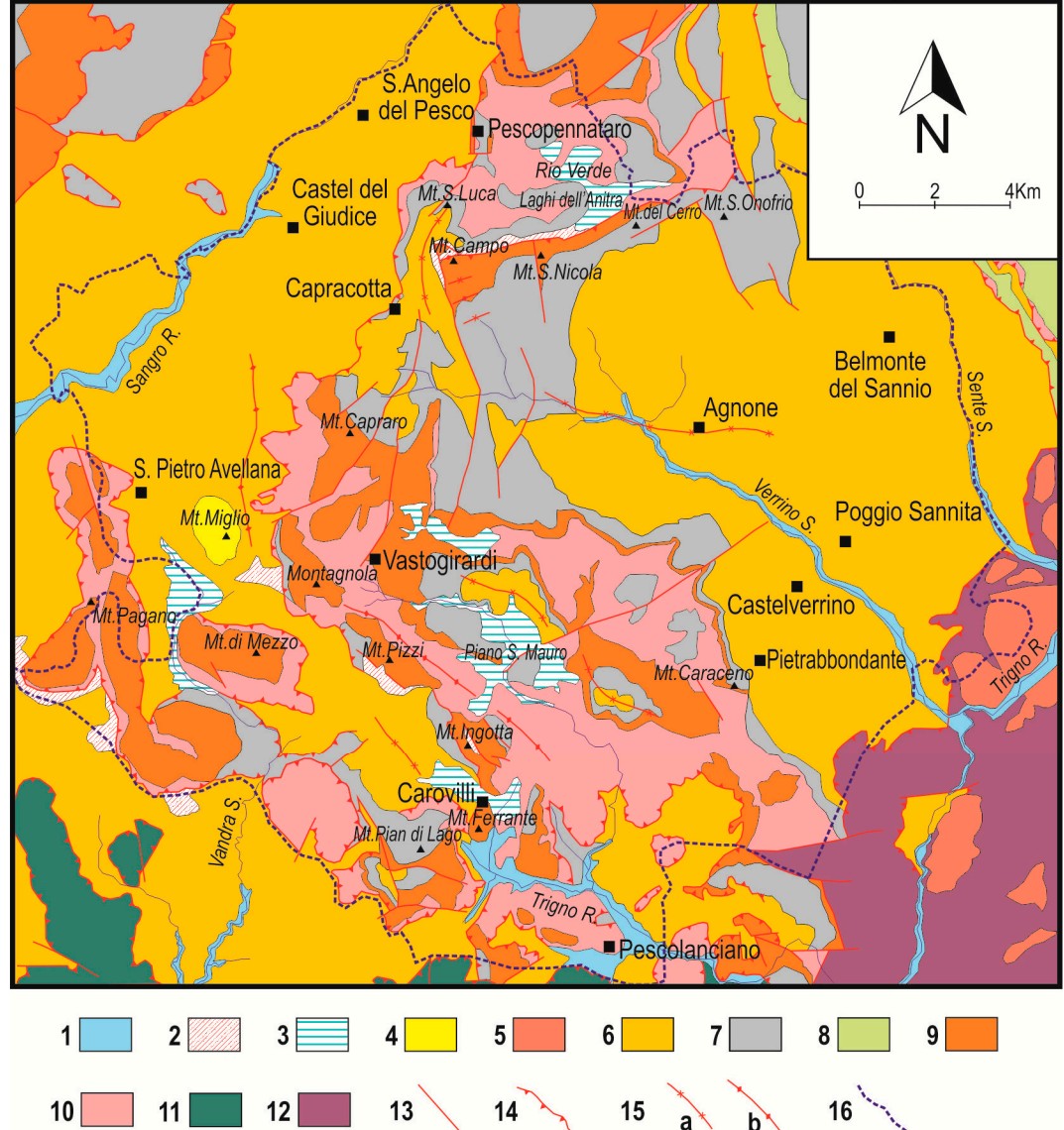

**Figure 2.** Geological sketch map of the Alto Molise area (modified from Vezzani et al. [62]). (1) Fluvial deposits (Holocene), (2) Slope debris (Late Pleistocene-Holocene), (3) Fluvial-palustrine deposits (Late Pleistocene-Holocene), (4) Limestones and polygenic conglomerates (Early-Middle Pliocene), (5) Siliciclastic deposits (S. Bartolomeo Flysch, Middle-Late Miocene), (6) Siliciclastic deposits (Agnone Flysch, Late Miocene); (7) Clayey marls and limestones (Marne ad Orbulina Formation, Late Miocene), (8) Marly limestones, marls and limestones (Tufillo Formation, Middle–Late Miocene), (9) Limestones and marls (Gamberale-Pizzoferrato Formation, Middle Miocene), (10) Varicolored clays and marls (Oligocene-Early Miocene), (11) Limestones and marly limestones (Frosolone Units, Late Cretaceous–Late Miocene), (12) Clays, marly clays and limestones (Sannio Units, Late Cretaceous–Early Miocene), (13) main faults, (14) main thrusts, (15) folds: a. syncline, b. anticline, 16) boundary of the Alto Molise macro-area.

The actual geological-structural setting of the Alto Molise area is the result of tectonics that acted from Late Miocene onwards. From the Messinian to the Middle Pliocene [63,64], the area was involved in thrusting that led to the tectonic juxtaposition of the Oligocene-Miocene stratigraphic units of the Molise basin on the Late Miocene Agnone Flysch. Then, from Late Pliocene to Early Pleistocene [61,63,64], the compressive structures were cut by strike slip and normal faults that acted from the Middle Pleistocene onwards according to a NE-SW direction of maximum extension.

As a result of this complex geological and tectonic history, the Alto Molise landscape is strongly dominated by structural landforms [20,57], especially monocline and anticline reliefs, often markedly asymmetrical, such as those forming Mt. Miglio, La Montagnola, Mt. Pizzi and Mt. Ingotta (Figure 2), typically aligned in the NW-SE direction. Slope processes and related landforms are widespread. Major phenomena are large rock falls with related talus slopes affecting the steep structural carbonate slopes (such as the one present along the western slope of Mt. Campo), together with complex landslides and phenomena of accelerated water erosion in the surrounding hilly areas. Where the tectonic juxtaposition of rigid carbonate rocks on plastic Miocene siliciclastic deposits has occurred, deep-seated gravitational deformations, as those affecting Mt. Pizzi, are also documented [65]. Furthermore, karst landforms are widespread where carbonate rocks crop out and are mainly represented by exokarst forms, such as karren and dolines, but also by some endokarst landforms, such as the Vomero Resurgence (Table 1).

From the bioclimatic point of view, the Alto Molise area is part of the temperate region [66], characterized by marked differences in winter and summer temperatures, precipitations concentrated in winter months, and summer aridity. Because of its climate conditions and geological-geomorphological features, the Alto Molise area is characterized by a high richness in fauna and flora, and related biodiversity. Its predominant forest vegetation is characterized by a high degree of naturalness [67], indicating that the evolution of the forest ecosystems is controlled especially by natural processes and only marginally influenced by human activities.

The high naturalistic value of the Alto Molise area is strengthened by the presence of numerous protected areas (Table 2) that occupy approximately 299.5 km$^2$, equal to 66% of its total surface area. Among these, a special mention deserves the Collemeluccio-Montedimezzo Alto Molise Unesco Man and Biosphere Reserve (Figure 1b), a large part of which falls in the Alto Molise territory.

**Table 2.** The Alto Molise natural protected areas.

| Site Type | Code and/or Name of Protected Natural Area | Surface (km$^2$) |
|---|---|---|
| EUAP | Riserva Naturale Orientata Collemeluccio | 4.22 |
| | Riserva Naturale Orientata Montedimezzo | 3.08 |
| non EUAP | Foresta Demaniale Regionale Bosco Pennataro | 3.45 |
| | Foresta Demaniale Regionale Monte Capraro | 1.95 |
| | Foresta Demaniale Regionale Bosco S. Martino e Cantalupo | 2.15 |
| | Oasi Legambiente Selva Castiglione | 3.00 |
| ZPS | IT7221132 Monte di Mezzo | 3.13 |
| | IT7221131 Bosco di Collemeluccio | 5.00 |
| SIC | IT7218213 Isola della Fonte della Luna | 8.67 |
| | IT7218217 Bosco Vallazzuna | 2.92 |
| | IT7211120 Torrente Verrino | 0.93 |
| | IT7212133 Torrente Tirino (Forra)–M. Ferrante | 1.45 |
| | IT7218215 Abeti Soprani–M. Campo–M. Castelbarone–Sorgenti del Verde | 30.33 |
| | IT7212134 Bosco di Collemeluccio–Selvapiana–Castiglione–La Cocuzza | 62.39 |
| | IT7212124 Bosco di M. di Mezzo–M. Miglio–Pennataro–M. Capraro–M. Cavallerizzo | 39.54 |
| MaB | Collemeluccio-Montedimezzo Alto Molise Biosphere Reserve | 252.68 [1] |

[1] The total surface of the MaB Reserve is indicated, including the portion that falls outside the Alto Molise macro-area.

From the cultural point of view, the Alto Molise area offers various tourist attractions, first some archaeological sites, such as the Temple-theater complex of Pietrabbondante and the Sanctuary of Vastogirardi (Figure 1b). There are also several villages and small towns with nice historical centres, like Agnone, famous for its craftmanship of bell casting, and Capracotta, well known for its cross-country skiing area. Furthermore, this area hosts rich evidence of agro-pastoral traditions that have contributed to the shaping of its cultural landscape, represented per excellence by the *thòlos*, characteristic stone shelters used by shepherds, and the *tratturi* (Figure 1b), i.e., ancient pastoral transhumance paths, also called drove roads [68]. Since the last decades, the tratturi have become

increasingly the subject of projects and studies [68,69] aimed at their recovery and fruition. Recently, they have been included in the national catalogue of historical rural landscapes [70,71], while the transhumance, the agropastoral practice of seasonal droving of livestock along migratory routes (tratturi) in the Mediterranean and in the Alps (Austria, Greece and Italy), was entered in 2019 in the representative list of the intangible cultural heritage of Humanity [72].

## 3. Materials and Methods

### 3.1. The Contents of the Application

The central contents of the proposed mobile application, hereinafter also called MoGeo App or simply App, are obviously the geosites and related geological itineraries.

The selection of geosites to be included in the App was based on the criteria safety, accessibility, scenic-aesthetic qualities, and interpretative potential that are in essence the selection criteria proposed by Brilha [73] for the qualitative assessment of sites suitable for geotourism use. Based on the data acquired during the Molise geosites inventory and other studies, the suitability of the 17 geosites was assessed by attributing a value for each criterion using scores (1—low; 3—medium; 5—high).

Safety and accessibility are considered indispensable criteria for geosite selection, to ensure their safe and unconditioned access and tourist use. For areal geosites, the accessibility and safety of both panoramic and on-site viewpoints were assessed.

The scenic-aesthetic qualities of geosites, i.e., the visual appeal and the natural beauty of a site, as well as the aesthetic qualities of the surrounding natural landscape, are considered of great importance to attract people to geosites [23,74–76]. Moreover, they can facilitate the interpretation of geosites, stimulate the curiosity of visitors and their desire to understand the geological-geomorphological features of geosites as well as the processes that underlie their genesis and evolution.

Despite the importance of aesthetics in tourism, according to Kirillova et al. [77], basic questions of tourist aesthetic judgment are still under-explored. These authors provide an important contribution to the understanding of "what makes a destination beautiful" by identifying and investigating nine themes and related dimensions of tourist aesthetic judgment in the context of both nature-based and urban tourist destinations. Another important aspect implicated in the aesthetic judgment is highlighted by Mikhailenko et al. [78] who propose a simple aesthetic-based classification of geological structures in outcrops based on their pattern, so as to take into account visions and attitudes of visitors and help evaluating the attractiveness of geosites.

To assess the scenic-aesthetic qualities of geosites, we have considered the following indicators: shape, vegetation, naturalness/anthropic modifications, chromatic variety and contrasts, and uniqueness. These indicators comprise some of the themes identified by Kirillova et al. [77], especially the shape. The latter, in particular, is closely related to the landscape setting and geological structure and includes the pattern sensu Mikhailenko et al. [78].

The interpretative potential of geosites is considered essential for disseminating geological information to non-geologists. It is closely related to the capacity of a geosite to be easily understood by law people and, therefore, to its representativeness, i.e., to its capacity to illustrate geological elements and processes.

The choice of other contents (Figure 3) to be included in MoGeo App was guided by the idea of enhancing and promoting not only geosites and related geo-itineraries, but also the overall naturalistic and cultural contexts of the selected areas. This is also in the awareness that the connection between natural and cultural heritage can represent a strength and a push factor for geotourism promotion, by offering richer and more varied experiences to visitors, who are perhaps not experts in geology or geology lovers, but simply families or lovers of landscape, nature and culture.

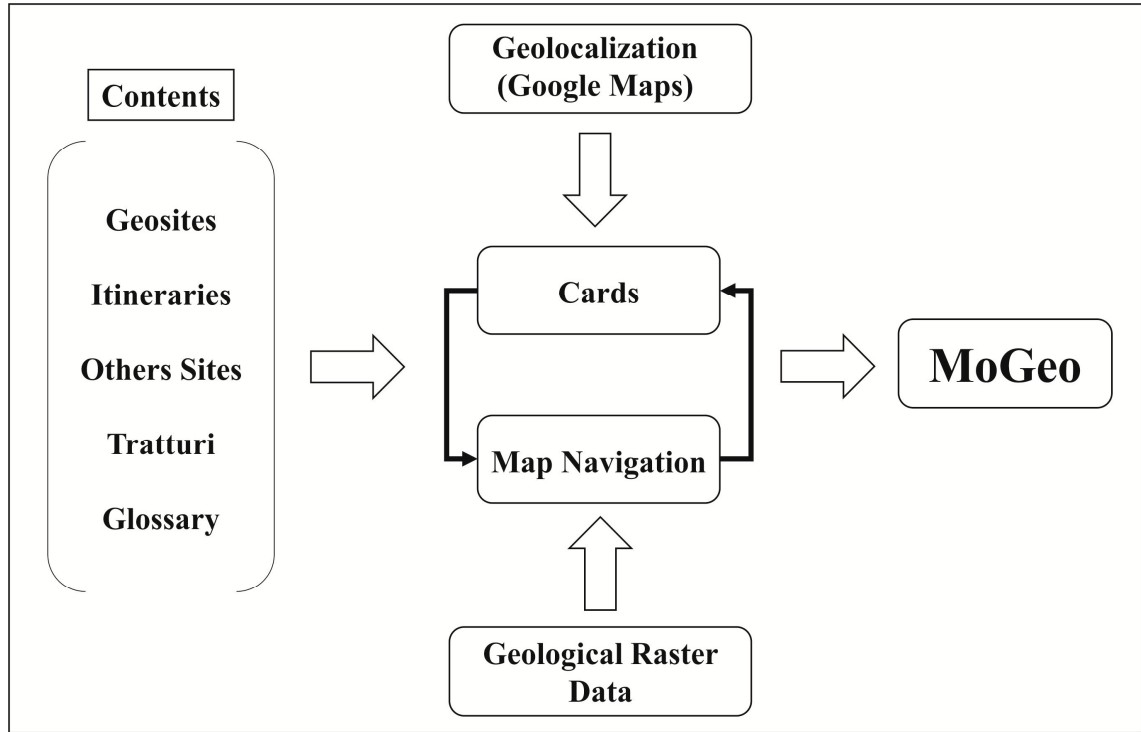

**Figure 3.** Flow Chart of the MoGeo App.

Therefore, the contents included in MoGeo App consist of:

- descriptive cards of selected geosites
- a glossary of scientific terms
- geological itineraries
- a simplified geo-lithological sketch map
- Other sites of cultural and/or naturalistic interest
- the tratturi

To allow the use of the App to both Italian and foreign tourists, all inserted titles and texts have been written in two languages, Italian and English.

For each selected geosite, a descriptive card was prepared that contains all essential information for understanding and for autonomously visiting the site. The inserted information was largely extracted from the Molise geosites inventory data archive [49,58] and other studies conducted about geosites/geotourism in the Alto Molise area [18,20,57].

Each card contains, in addition to the description of the geosite, information on the main geoscience interests. To select the main geoscience interests to insert in the cards, we did not simply consider the major scientific interests of geosites (Figure 1c and Table 1), but geological themes that could be of wider and perhaps environmental interest. For example, we considered the geoscience interest "Landscape instability" (see section results and Table 3), which is closely linked to the issues of natural hazard and risk and, in turn, to global issues such as the climate change. To ease the disclosure and interpretation of geosites, the descriptive cards were enriched with specific illustrative material (photos, sections, geological sketches, 3D schemes, etc.). Furthermore, to allow an optimal appreciation of each geosite, the best on-site and panoramic viewpoints are indicated.

**Table 3.** List of selected geosites. ID, scores obtained for each criterion, type of proposed observation points, main geoscience interests, and other natural and cultural interests. H = Hydrogeology, Se = Selective erosion, L = Landscape instability, Pc = Paleoclimate; St = Stratigraphy, Sed = Sedimentology, T = Tectonics, Pe = Paleoenvironment, Pa = Paleontology.

| ID | S [1] | A [2] | SE [3] | IP [4] | O [5] | P [6] | MGI [7] | Other Interests |
|----|-----|-----|------|------|-----|-----|-------|-----------------|
| D1 | 5 | 3 | 3 | 5 | X | | H | Flora, vegetation and fauna |
| D2 | 5 | 5 | 5 | 5 | | X | Se–H | Flora, vegetation and fauna—Human history |
| D3 | 5 | 5 | 5 | 3 | X | X | Se–L | Flora, vegetation and fauna |
| D4 | 5 | 5 | 5 | 5 | | X | Se–Pc | Flora, vegetation and fauna |
| D5 | 5 | 5 | 5 | 5 | X | | H | Flora, vegetation and fauna—Human history |
| D6 | 5 | 5 | 3 | 5 | X | | Se–St | Flora, vegetation and fauna—Human history |
| D7 | 5 | 5 | 5 | 5 | X | X | Se–L–St–Sed | Flora, vegetation and fauna—Agro-pastoral tradition |
| D8 | 5 | 5 | 3 | 5 | X | X | St–Sed | |
| D9 | 5 | 5 | 5 | 3 | X | X | Se–St–Sed–T | |
| D10 | 5 | 3 | 5 | 5 | X | X | Se–Pe–Pa | Flora, vegetation and fauna |
| D12 | 5 | 5 | 5 | 5 | X | X | Se–St–Sed | |
| D13 | 5 | 5 | 5 | 5 | X | X | Se–T–Pa | |
| D14 | 5 | 3 | 5 | 5 | X | X | H | Flora, vegetation and fauna |
| D15 | 5 | 5 | 3 | 5 | X | X | Se–St–Sed | Human history, Architecture and Handicraft |

[1] Safety; [2] Accessibility; [3] Scenic-aesthetic qualities; [4] Interpretative potential; [5] On-site view; [6] Panoramic view; [7] Main geoscience interests.

We tried to use the simplest language possible to be clear even for non-geologists. However, to safeguard the scientific rigor, it was not possible to exclude certain scientific terms, so we included among the contents a glossary of scientific terms.

Considering that the use of geological itineraries and viewpoints is an important tool in geotourism activities [23,79], we have created also some itineraries (four at the moment). These itineraries allow the joint visit of different geosites and involve all selected geosites. Furthermore, to attract a large audience, they follow the main road system. For each itinerary, we have detailed the path and the sequence and location of stops. Based on logistic conditions and facilities (presence of parking areas, rest areas, etc.), the best on-site and panoramic viewpoints were selected. For all stops, descriptive cards were drawn up.

Panoramic viewpoints become particularly important as they allow the observation of sites represented by large landforms, which can be best appreciated from a distance. The importance of panoramic viewpoints, especially where on-site views of geosites are not useful or available has led to conceptualize a specific category of geosites: the viewpoint geosites [80,81]. According to these authors, the location of panoramic viewpoints has to take into account not only the quality of the site view they allow (clarity of features, good light, good visibility, etc.), but also the environmental context surrounding the target, as well as the conditions and pattern of the standpoint.

In the choice of our panoramic viewpoints, also some of the aforementioned criteria for viewpoint geosites have been considered. So we selected standpoints that satisfy the following conditions: easily accessible (normally located along the main road), not placed in private properties, with good safety conditions, with null to minimum anthropogenic degradation, preferentially inserted in a context characterized by few human interventions and a medium to high degree of naturalness, not covered by dense vegetation, and allowing the view of geological/geomorphological features that in many cases stand out (for color contrasts, vertical elevation, etc.) from the surrounding landscape. Furthermore, where possible, we selected several panoramic points for the same geosite to allow the specific observation of different/separate elements of the geosite. The preferential location of panoramic viewpoints along main roads was also guided by the need to guarantee the mobile phone coverage.

To best explain in the application the geosites from these panoramic views, we prepared panoramic photos clearly visible at the screen size, partially "retouched" to better put in evidence the features to be observed, and simplified sketches to illustrate the geological setting and the relief features.

To sustain geological information given in the cards, we realized a geolithological sketch map in the GIS environment, mainly based on the geological data extracted from the Geological Map of

Molise in scale 1:100,000 [62]. Data on tectonics features, which may be excessively complex for an audience of non-geologists, were not included in the map.

The other sites included in the App were selected by considering not only the best known cultural and natural sites/areas of Alto Molise, but also lesser known sites, to give a comprehensive overview of landscape, flora, vegetation and fauna, traditions, history, and archaeology of this territory. A simple and concise information card was produced for each of these sites containing, where available, the link to the official webpage of the site for further information.

Finally, the traces of the three tratturi that cross the Alto Molise territory were inserted in the App, as they represent important and distinctive landscape elements. These traces were extracted from the GIS project implemented during the Molise regional geosites inventory [49]. A general presentation of this theme and a descriptive card for each tratturo were prepared.

*3.2. The Implementation of the Application*

To develop our App, we considered the following expected characteristics and performance requirements:

- The application should be of easy and linear use. Contents should be accessible both by following a drop-down menu and by starting from an interactive map;
- All sites and itinerary stops, as well as the position of the user, should be geolocalized on an interactive map;
- The application should be usable on any kind of Android device, such as smartphones with varying screen sizes and tablet computers;
- The application should be fast and should not require a high storage capacity of mobile devises;
- The application should be easily updatable and expandable, by adding features such as tools or other data.

MoGeo App has been designed as a hybrid mobile app [45], a combination of web and native mobile applications, in which the cartographic part interfaces directly with the host operating system (in our case Android), while the information cards are inserted on a remote responsive website, optimized for mobile. The web pages are loaded via WebView, a native component, and displayed on the device as if they were themselves native. The advantage of this approach is to view the contents both from MoGeo App and directly from the Web Browser by connecting to the specific website. Through the geolocalization, it is possible to know the position both of the user and the single site, the distance between them, and the shortest way to reach the site.

In detail, MoGeo App resides in two different virtual spaces:

- The cartographic part is developed through the Android Studio Software. This is a fully integrated development system, created and made freely available by Google [82] for the development of new applications. In particular, three different map types are used (Google street and satellite, and geological maps) to localize the geo-touristic data related to different information cards (Figure 3).
- The other part includes the cards containing the information collected for all contents (Figure 3). All information has been transcribed in HTML format by using a Content Management System (CMS), called WordPress, and has been stored on an Italian Web Platform that AlterVista [83] makes freely available for all users registered on this platform.

To avoid speed and performance reductions of the application, information cannot be stored on the mobile devise, but is simply uploaded by using an Internet connection.

**4. Results**

*4.1. Characteristics of Selected Geosites*

Fourteen of the 17 Alto Molise geosites have been included in the App (Table 3), based on scores obtained for selection criteria safety, accessibility, scenic-aesthetic qualities and interpretative potential.

Geosites D11, D16 and D17 (Table 1) were excluded due to their medium to low scores in safety and accessibility: geosite D11 due to medium scores both in safety and accessibility, and geosites D16 and D17 (respectively a karst cave whose access is restricted to experts with caving equipment and a landslide area, see Table 1) due to their low and medium scores respectively in accessibility and safety.

The 14 selected geosites have achieved a high score in safety as they can be observed in complete security conditions, with nil to minimum risk for visitors. Nine geosites can be appreciated by both on-site and panoramic views (Table 3), three and two, instead, only by on-site and panoramic views, respectively.

None of the selected geosites present any use limitation due to access permissions. Furthermore, most of the selected geosites have obtained a high score in accessibility because they do not have access difficulties and are easily reached by paved roads. The geosites Capo di Vandra spring and Verrino Stream waterfalls (D1 and D14 in Table 3) are characterized only by medium scores in accessibility, because the first site can only be reached by completing the last part of the route by foot or with an off-road vehicle, while the second site, which is located within the Verrino Fluvial Park, can only be reached on foot. In addition, a medium store in accessibility marks the geosite Mt. Ingotta anticline (D10 in Table 3) as it is necessary to walk a path on foot to reach the on-site view for observing in detail the fossiliferous strata.

All selected geosites, except two, got a high score in interpretative potential, as they can be easily understood by a large audience, in particular also by lay people without a geological background. The two geosites with medium scores in interpretative potential (D3 and D9, Table 3) were anyway selected because they are characterized by high scenic-aesthetic qualities.

Finally, most of the selected geosites have achieved a high score in scenic-aesthetic qualities, while four of them (D1, D6, D8 and D15) reached a medium score, but were selected because of their high interpretative potential.

The geoscience interests of selected geosites range between Hydrogeology and Landscape instability (Table 3), with the latter being the most represented. Other interests of geosites are Flora, vegetation and fauna, Agro-pastoral tradition, Human history, Architecture, and Handicraft (Table 3).

### 4.2. How to Access Information Using MoGeo App

MoGeo App, which is downloadable by using the link https://geositi.altervista.org/download or the QR code in Figure 4, offers a simple and rapid way to reach information about contents included. Once you open the Homepage (Step 1, Figure 4), information can be accessed through two separate ways: the drop-down menu and the interactive map.

By staying on the Homepage, a drop-down menu can be opened that allows consulting the list of contents (Step 2, Figure 4). By making a selection on one of these contents, for example geosites, the user is redirected to the relative list of geosites (Step 3, Figure 4). By making a further selection on the latter, it is possible to reach the page of the specific site. In addition, at the top of the map view there are three buttons that allow respectively to return to the Homepage, to open the legend and to switch to satellite mode.

Starting from the interactive map, information can be accessed through the sites/places of interest that are localized on it by means of classic Google Maps textures and indicated with different colors according to the type of content (Step 2, Figure 4). By clicking on one of them, a popup opens that contains first information about the site, such as a photo and the name (Step 3, Figure 4). By clicking on the latter, the user is redirected to the information card that contains the description and illustration (text, photos, schemes, etc.) prepared for the specific site (Step 4, Figure 4).

### 4.3. Contents of MoGeo App

The content Geosites includes a general presentation and the information cards prepared for the 14 geosites. The presentation card (Figure 5) provides basic notions about the meaning of the term geosite and a short information on the Alto Molise geosites. The information cards provide information

about the origin, geological-geomorphological features, and main geoscience interests of the geosite, as well as about the age of rock formations involved. All cards are enriched with illustrations, especially photos (Figure 5), geological sections and 3D schemes. The latter have been included mainly to explain some specific geological features such as various types of faults (Figure 5).

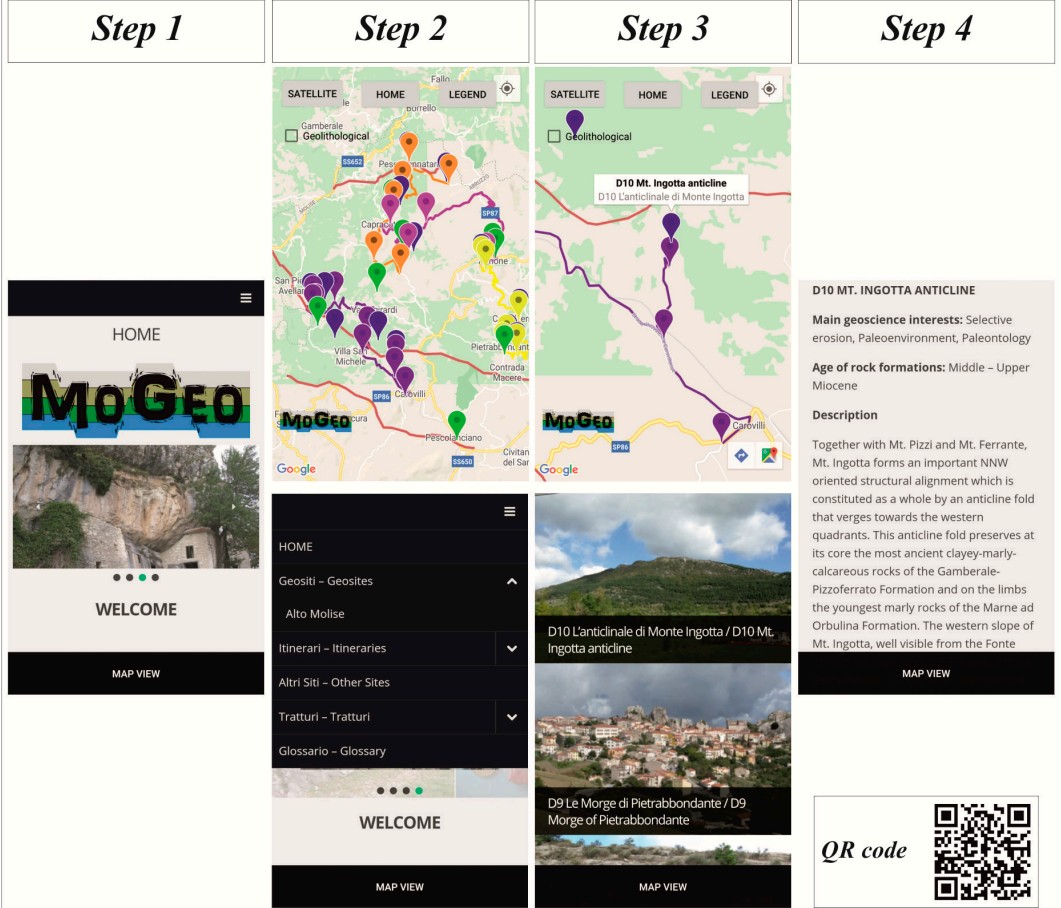

**Figure 4.** The main structure of the MoGeo App. Step 1: the Homepage that addresses to the two types of data access; Step 2: the interactive map and the drop-down menu access; Step 3: the opening of a pop-up after clicking a site on the interactive map, and the opening of a specific list of sites by selecting a single content from the drop-down menu; Step 4: the information card of a geosite. The figure contains also the QR code to download the MoGeo App.

The content Itineraries give access to the list of created geological itineraries (I1–I4, Table 4 and Figure 6), a card that provides a short information on each itinerary, and the descriptive cards that illustrate each single stop. Two of these itineraries (I1 and I2) mainly develop in the northern sector of the Alto Molise area, the other two (I3 and I4) mainly in the southern sector. They are made of a variable number of stops, from a minimum of 5 (I2, Table 4) to a maximum of 9 stops (I4, Table 4), and embrace several geoscience interests (Table 4) that allow visitors to deepen certain geological topics. Stops are mostly very easy to access, as the itineraries run largely along the main roads. Walking paths are needed only in some cases, precisely to reach the first stop of I2 itinerary, located within the Verrino Fluvial Park, and the third and ninth stops of I4 itinerary (Table 4). Itineraries I1, I2 and I4 partly cross protected natural areas (Table 4).

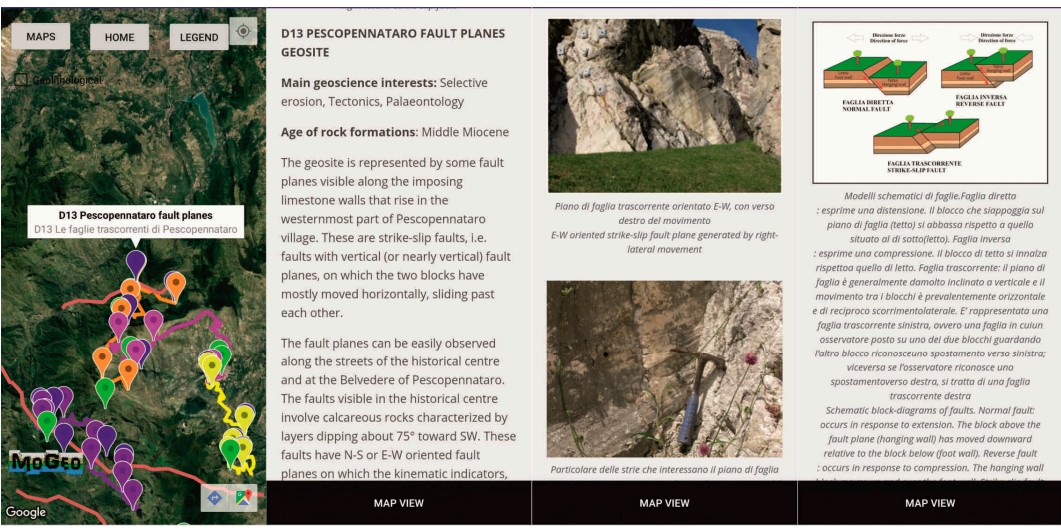

**Figure 5.** Sequence of screenshots extracted from MoGeo App that illustrate various aspects of the content Geosites.

**Table 4.** Main characteristics of the created four itineraries.

| Itinerary | Stop | Geosite | O [1] | P [2] | Geoscience Interests | Route | Natural Areas |
|---|---|---|---|---|---|---|---|
| I1 Capracotta-Rio Verde Springs | 1 | D7 | | X | Selective erosion Tectonics Landscape instability Paleoenvironment Hydrogeology | Main roads | IT7218215 SIC |
| | 2 | D7 | | X | | | |
| | 3 | D7 | X | X | | | |
| | 4 | D6 | X | | | | |
| | 5 | D13 | X | | | | |
| | 6 | D5 | X | | | | |
| I2 Capracotta-Agnone | 1 | D14 | X | | Selective erosion Landscape instability Hydrogeology Stratigraphy Sedimentology | Path Main roads | Verrino Fluvial Park |
| | 2 | D7 | X | | | | |
| | 3 | D7 | | X | | | |
| | 4 | D7 | | X | | | |
| | 5 | D15 | X | | | | |
| I3 Pietrabbondante-Agnone | 1 | D9 | | X | Stratigraphy Sedimentology | Main roads | |
| | 2 | D9 | X | | | | |
| | 3 | D8 | X | | | | |
| | 4 | D12 | X | | | | |
| | 5 | D15 | | X | | | |
| | 6 | D15 | X | | | | |
| I4 Carovilli-Capo di Vandra | 1 | D10 | | X | Selective erosion Landscape instability Paleoclimate Paleoenvironment Hydrogeology | Main roads Paths | MAB Reserve Collemeluccio-Montedimezzo Alto Molise |
| | | D3 | | X | | | |
| | 2 | D10 | | X | | | |
| | 3 | D10 | X | | | | |
| | 4 | D3 | | X | | | |
| | 5 | D3 | | X | | | |
| | 6 | D2 | | X | | | |
| | 7 | D2 | | X | | | |
| | 8 | D4 | | X | | | |
| | 9 | D1 | X | | | | |

[1] On-site view; [2] Panoramic view.

Regarding the content Other sites (Figure 7), we have included for now nine sites, three of naturalistic and six of cultural interest (Figure 1), which are well distributed throughout the Alto Molise territory. The sites of naturalistic interest are the Garden of Apennine flora of Capracotta, the Collemeluccio–Montedimezzo Alto Molise MaB Reserve and the Verrino Fluvial Park. The sites of cultural interest are the Samnite Temple of Vastogirardi, the Temple-theater complex of Pietrabbondante, the Bell Museum of Agnone, the Stone Museum of Pescopennataro, the Castel of Pescolanciano, and the Museum of Transhumance of Agnone. Information cards illustrate with a synthetic text and some photos major features of each site (Figure 7).

Regarding the content Tratturi, all three major drove roads that cross the Alto Molise territory, the tratturi Ateleta–Biferno, Castel di Sangro–Lucera and Celano–Foggia (Figure 7), were included in our App. The traces of the first are visible in the northern sector of the Alto Molise area, while the traces of the other two are preserved in the southern sector. Also for the tratturi, in addition to the

single information cards (Figure 7), a general presentation card was compiled. This card contains information on the transhumance and related paths, i.e., the tratturi, and allows to have an overview about this ancient agro-pastoral practice and all major tratturi that cross the Molise territory.

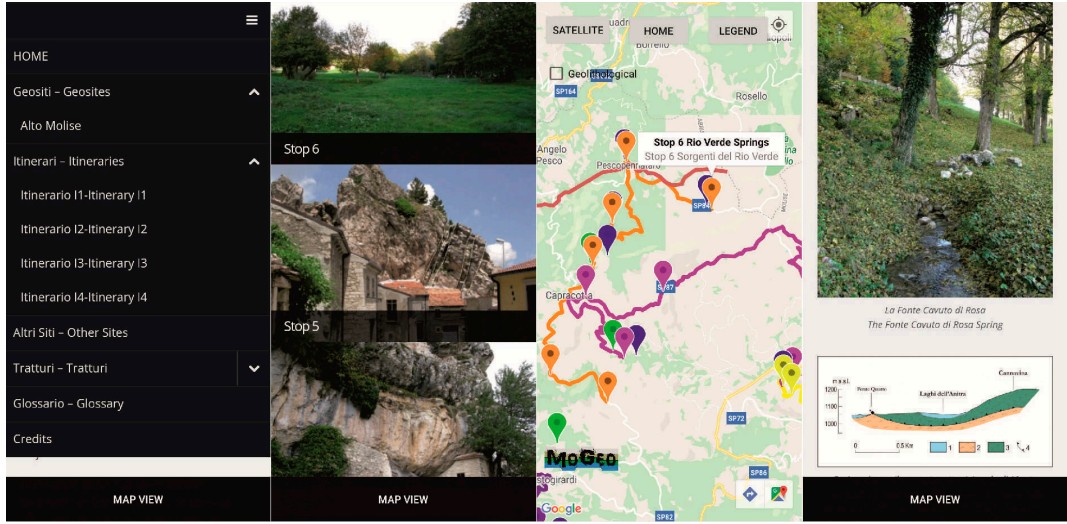

**Figure 6.** Sequence of screenshots extracted from MoGeo App that illustrate various aspects of the content Itineraries.

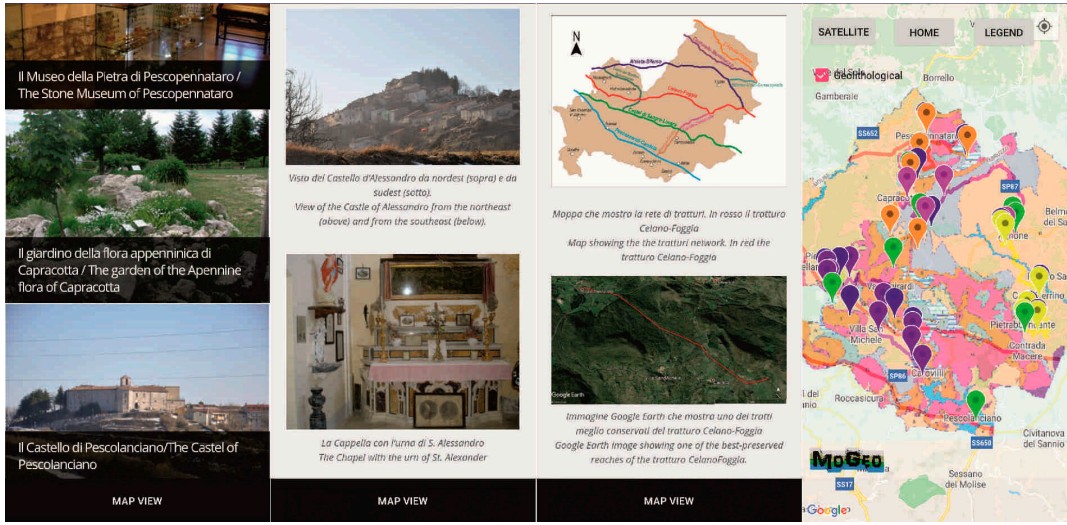

**Figure 7.** Sequence of screenshots extracted from MoGeo App that illustrate various aspects related to contents Other sites, Tratturi and Geo-lithological map.

Two further products were included in MoGeo App: A geolithological map and a glossary of scientific terms. The geolithological map (Figure 7) allows visualizing each site in its specific geological context, contributing to a better understanding of the geological information provided by the cards. The glossary of scientific terms, instead, can help not only to better understand information provided by cards for geosites, but also represents a useful tool for tourists who wish to deepen their knowledge about geological elements/topics. It allows easy research on specific scientific terms included in the information cards.

## 5. Discussion and Conclusions

MoGeo App represents a smart tool that allows the access to specific geotouristic information about sites/places of geological, natural and cultural interest geolocalized on Google earth maps and satellite views, by using a small mobile device, such as a smartphone.

It offers a simple and rapid way to reach information and, thanks to its hybrid nature, can be defined as a user-friendly tool, because of the velocity with which it elaborates the commands of the user and the accuracy of its geolocation functionality. In fact, this App has been developed considering that not all users have sophisticated and high-performing mobile devises and/or high knowledge and manual skills in using their smartphone. Obviously, some difficulty or limitation can arise from the diversity of mobile devices and related graphic performance capabilities.

In addition, MoGeo App is based on an archive of information that can be updated, modified and enriched with new information and cartographic tools. In this case, the only limitation of the information that can be accessed is related to the computing power of the device.

The developed application is suitable for guiding individual visits (perhaps even involving only single sites) and tours that are selected among the offered itineraries or developed by users according to their specific geo-naturalistic and cultural interests and time available. Therefore, it is able to meet the needs and preferences of a large audience, including families as well as hikers, amateur geologists and admirers of natural beauty, encouraging them to visit these places often forgotten by the media.

Besides its use during visits/excursions, MoGeo App can also be useful for analyzing preventively the areas of interest, especially for defining the best road connections and acquiring information about the geographical and geological contexts that characterize the area and the individual sites that can be visited. In Italy, there are some examples of applications created for geotourism purposes in other regional areas (e.g., [84,85]). By comparing MoGeo App with other applications, it is possible to observe some differences. One of the main differences is that MoGeo allows to process together different types of information, organized in separate categories, not just geosite information. In fact, great importance assumes the possibility of accessing with a single application different contents (geological, natural, cultural, etc.) that can be geolocated on interactive maps, by using mapping service portals such as Google Maps. Also the drop-down menu appears extremely useful, as it allows to reach the different contents without using the map. In this way, the individual user can choose the way to receive information according to his needs.

Other strengths of this application are its light and smart structure. In particular, the information is acquired directly remotely without downloading data to the smartphone. This allows to avoid speed and performance reductions of the application.

However, MoGeo App also has some limitations. Being developed for smartphones/tablets, due to the reduced size of the screens, the scale of images cannot be of elevated detail and resolution. Another limitation is that, for now, MoGeo does not include the use of Augmented Reality (AR), which could be developed in the future.

To create our application, we chose to use only the Android operating system, because it is the most widespread in Europe and in Italy (in December 2019, 81.4% of Italian smartphone users are Android [86]) and provides for free both the possibility to publish the application on its store and to use the API (Application Programming Interface) to connect to the tools of Google Maps. However, in the future we would like to create a version for IOS, so as to reach all smartphone users.

During the phase of design, MoGeo App has been tested remotely and directly in the field on a group of a dozen people without geological background to get a first quick feedback on the ease of use and efficiency of the application, as well as on the curiosity towards and liking of proposed contents. By questioning the group used as for the test, it was possible to verify a large appreciation of contents and a positive judgment about the lightness of the application (~3 Mb), which allows a quick download on the mobile phone, even on less technologically advanced smartphones. Furthermore, the logic and structure of the application resulted in being intuitive for everyone. In particular, no one had difficulties in finding information both on the geosites and the territory in general, managing to

create a personal visit itinerary. There was a widespread appreciation of the cards for the clarity and simplicity of contents. The possibility of a combined consultation of different contents (cards) was also judged positively. In the field, the application works quite well and the function that allows to reach a given site using the Google Map tools incorporated in MoGeo has been particularly appreciated. Only in a few cases, reception problems were found due to the local weakness or lack of the cell phone signal, above all in the high mountain areas. Wind and Iliad network users had more problems than TIM and Vodafone network users.

Further surveys on the reception and use of MoGeo App can certainly help improve it and are scheduled. In the future, we intend to conduct more systematic tests involving a group of families, to better consider the needs and interests of this specific target group and try to shorten the distance between contents for experts and lay people. This will be done by collecting through questionnaires the interest, appreciation and/or criticism expressed by families, as well as the difficulties encountered in using the App and understanding of contents; in short, all the useful tips to improve the App.

The contents of MoGeo App are specifically designed for geotourism purposes, but its structure can be used for promoting and disseminating geoheritage to different target groups, by adapting the cards' content and the language. For example, contents for educational targets or dedicated only to children/teenagers (considering that digital tools help attract and engage this kind of audience) can be devised.

In conclusion, MoGeo App appears to be a flexible, updatable digital tool that can support different contents and can be adapted to various target groups and to other regional contexts, both inside and outside of Italy.

**Download MoGeo App:** https://geositi.altervista.org/download or QR code in Figure 4.

**Author Contributions:** Conceptualization, G.D.P., F.F. and C.M.R.; methodology, G.D.P., F.F., L.M. and C.M.R.; software, L.M.; validation, G.D.P., F.F.; investigation, F.F.; data curation, F.F. and C.M.R.; writing—original draft preparation, G.D.P., F.F., L.M. and C.M.R.; writing—review and editing, G.D.P. and C.M.R. All authors have read and agreed to the published version of the manuscript.

**Funding:** This research received no external funding.

**Acknowledgments:** The authors wish to thank the anonymous reviewers whose suggestions greatly improved the manuscript.

**Conflicts of Interest:** The authors declare no conflict of interest.

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
