# Peer review of "MoGeo, a Mobile Application to Promote Geotourism in Molise Region (Southern Italy)"

_resources, doi:10.3390/resources9030031_

Round 1

Reviewer 1 Report

Dear Authors,

Thank you for the interesting contribution. Certainly, the number of geotourism products, such as the mobile application in question, is insufficient.

I have some questions and suggestions related to the manuscript:
1. I think it would be worth doing a brief review of existing geotourism applications. Indicate their strengths and weaknesses.
2. Please add geographical coordinates to Figure 1. The map of Italy could be slightly larger.
3. Please explain the term "tratturi".
4. The application cannot be installed because a password is required.

5. Please provide basic information about the geotourism in the region described in the paper. Does any geotouristic products (leaflets, trails, guides) exists there?

Author Response

Authors: We thank the reviewer for his valuable suggestions. All the manuscript corrections are reported below, describing them point by point.

Point 1: Dear Authors,

Thank you for the interesting contribution. Certainly, the number of geotourism products, such as the mobile application in question, is insufficient.

I have some questions and suggestions related to the manuscript:

Response 1: We thank the reviewer for all his suggestions.

Point 2: I think it would be worth doing a brief review of existing geotourism applications. Indicate their strengths and weaknesses.

Response 2: We have added a brief review in section Introduction (lines 66-73).

Point 3: Please add geographical coordinates to Figure 1. The map of Italy could be slightly larger.

Response 3: The geographical coordinates have been added and the map of Italy has been enlarged.

Pont 4: Please explain the term "tratturi".

Response 4: The term tratturi is already explained in section 2 (lines 196-197).

Point 5: The application cannot be installed because a password is required.

Response 5:  We apologize about that. We had placed the password in the cover letter. However, the password to install the application is: 1234.

Point 6: Please provide basic information about the geotourism in the region described in the paper. Does any geotouristic products (leaflets, trails, guides) exists there?

Response 6: This have been done. Unfortunately, geotouristic products are very few. A short review of all products that regard or include in some way geosites and their promotion have been inserted (lines 80-95).

Reviewer 2 Report

MoGeo, a Mobile Application to promote Geotourism in Molise Region (southern Italy)

by Francesca Filocamo et al. submitted to Resources journal

In general, the manuscript is well organized, with adequate references regarding geotourism and similar approaches on technological tools to support geotourism in the field.

However, I consider that the work lack some innovation in these topics and could be more complete regarding results of the MoGeo app with geotourism purposes.

1. Selection of sites and their assessment

The authors refer that “The selection of geosites was based on the criteria safety, accessibility, scenic-aesthetic qualities and interpretative potential that are substantially the selection criteria proposed by [59] for the qualitative assessment of sites suitable for geotourism use. Based on the data acquired during the Molise geosites inventory and other studies, the suitability of the 17 geosites was assessed by attributing a value for each criterion using scores (1-low; 3-medium; 5-high) (lines 177-181). These results are presented in Table 1.

It seems that information on the 17 sites selection is missing (how these 17 sites were selected?; with which criteria?). Besides, the criteria used for geotourism use are few and some of them very subjective (scenic-aesthetic qualities and interpretative potential) . Other methods include criteria like “visibility”, “other uses on the site”, “other interests”, “distance to support infrastructures (interpretative centers, restaurants, toilets, etc.)”, “legal status (because of possible restrictions)”, which are way more objective and suitable for this kind of assessments.

Nevertheless, this is not the main topic of the work, thus the main remark is related with the MoGeo app.

2. MoGeo app

The authors refer that the knowledge already available makes the Alto Molise area a good starting point to develop and test the application (line 83). In my opinion, only the development task is presented.

It is a pity that the results do not include the use of the app by real tourists and their assessment of it. It would be very important to know more about the real efficiency of the tools provided by the app and mostly about the contents legibility  by the users.

In chapter 5 ( Discussion and Conclusions, lines 424-432) the authors say that “During the phase of design, MoGeo App has been tested remotely and directly in the field on a group of people to get a first quick feedback on the usability of the App system and on the curiosity towards and liking of proposed contents. These tests have evidenced a broad appreciation of the App but also possible local difficulties related to the access of the telephone network. Regarding foreign tourists, this problem will probably be easily overcome by the possibility to change the mobile network operator in roaming conditions. For Italian tourists, instead, this solution may be not applicable, so a future implementation of the application is actually being evaluated to favour the off-line use of contents by allowing their download at least for areas with ascertained difficulty of internet access”. 

They even recognize that the results would be more complete if tests were performed with tourists (lines 433-439): Further surveys on the reception and use of the App can certainly help improve it and are scheduled. In the next future we intend to conduct more systematic tests involving a group of families, to better consider the needs and interests of this specific target group and try to shorten the distance between contents for experts and lay people. This will be done by collecting through questionnaires the interest, appreciation and/or criticism expressed by the families, as well as the difficulties encountered in using the App and understanding of contents; in short, all the useful tips to improve the App. 

Also because of that my opinion is that this work is incomplete and its final publication would benefit of results from a survey conducted with tourists, what would be the real test of the application.

I consider also that the application could incorporate some innovative features that are being used and applied in the last years like Augmented Reality (AR) applied to geological crops and elements.

Besides, the app is limited to Android users, which eliminate a high quantity of tourists that use Apple devices, and it seems to be internet availability dependent at the sites.

These issues (that I consider as weaknesses) should be at least incorporated in the discussion, recognizing the need to improve the tool. By the other hand, these are weaknesses that surely will be noticed by users in the field, what reinforce the importance of such component in the presented work. 

Other minor remarks:

- Are the app contents only in Italian? Par example, in Fig. 6 (line 372-373) we can see the menu in English and the contents in Italian.

- Are the texts inside the figures with app print screens legible? 

Author Response

Authors: We thank the reviewer for his valuable suggestions. All the manuscript corrections are reported below, describing them point by point.

Point 1: MoGeo, a Mobile Application to promote Geotourism in Molise Region (southern Italy) by Francesca Filocamo et al. submitted to Resources journal.
In general, the manuscript is well organized, with adequate references regarding geotourism and similar approaches on technological tools to support geotourism in the field.
However, I consider that the work lack some innovation in these topics and could be more complete regarding results of the MoGeo app with geotourism purposes.

Response 1: We thank the reviewer for all his suggestions.

Point 2: Selection of sites and their assessment
The authors refer that “The selection of geosites was based on the criteria safety, accessibility, scenic-aesthetic qualities and interpretative potential that are substantially the selection criteria proposed by [59] for the qualitative assessment of sites suitable for geotourism use. Based on the data acquired during the Molise geosites inventory and other studies, the suitability of the 17 geosites was assessed by attributing a value for each criterion using scores (1-low; 3-medium; 5-high) (lines 177-181). These results are presented in Table 1.
It seems that information on the 17 sites selection is missing (how these 17 sites were selected?; with which criteria?).

Response 2: The 17 geosites of Alto Molise area are part of the 100 geosites surveyed in Molise region during the regional geosites inventory carried out during the first half of the 2010s by the Department of Biosciences and Territory of the University of Molise in partnership and on behalf of the Molise Region. Geosite assessment has been based on a quantitative assessment method. In particular, to ascertain the scientific value of geosites, the geosites evaluation model developed by the Lazio Regional Park Agency in collaboration with ISPRA (and in coherence with the ISPRA’s national inventory geosite card) was adopted with some minor changes. This method is based on some of the major criteria used worldwide, especially: rarity, representativeness, and integrity/present conservation status.
Once calculated the representativeness, rarity, and scenic-aesthetic, historical-archaeological-cultural, and vulnerability values of geosites, the so-called “intrinsic value of the site of geological interest” (IVSGI) has been calculated, resulting from the weighted sum of representativeness, rarity, and scenic-aesthetic values.
To clarify this issue, missing information has been added in Section 2 (lines 112-119).

Point 3: Besides, the criteria used for geotourism use are few and some of them very subjective (scenic-aesthetic qualities and interpretative potential). Other methods include criteria like “visibility”, “other uses on the site”, “other interests”, “distance to support infrastructures (interpretative centers, restaurants, toilets, etc.)”, “legal status (because of possible restrictions)”, which are way more objective and suitable for this kind of assessments. Nevertheless, this is not the main topic of the work, thus the main remark is related with the MoGeo app.

Response 3: We agree, the used criteria are relatively few and certainly there may be a certain risk of subjectivity. However, we consider scenic-aesthetic qualities and interpretative potential two fundamental criteria that need to be evaluated for geotourism purposes, as evidenced by several works (e.g., Brilha 2016, 2018 and references therein). In particular, scenic-aesthetic qualities are considered significant initial factors for attracting casual geotourists to geosites (e.g., Chylińska 2019, Hose, T.A. 2010, and Gordon, J. E. 2018 and references therein).
In addition, some of the criteria mentioned by the reviewer, such as “visibility” and “legal status (because of possible restrictions)” are incorporated within the 4 criteria we considered. Visibility was considered in attributing the score to the “interpretative potential” criterion that is based on the representativeness value of the geosite, the visibility and the plurality of interests. Furthermore, the “legal status” of the site was considered in the attribution of the score to the "accessibility" criterion.
The “other interests” were anyway considered by us in terms of added value (see table 3) and most of the selected geosites are characterized by the association with other values.
To minimize the subjectivity of our assessments, we used the data archived in the Molise geosite inventory database (census cards of the 17 geosites) which is based, as mentioned above, on a quantitative assessment model.
To better clarify this issue we have made some changes/additions in section 3.1 (lines 222–234), also taking into account the suggestions of Reviewer 2, and section 4.1 (line 362).

Point 4: MoGeo app
The authors refer that the knowledge already available makes the Alto Molise area a good starting point to develop and test the application (line 83). In my opinion, only the development task is presented.

Response 4: We agree. Essentially, only the development task is presented (we have corrected this part of the text). Only first, expeditious tests have been carried out (please see the next point).

Point 5: It is a pity that the results do not include the use of the app by real tourists and their assessment of it. It would be very important to know more about the real efficiency of the tools provided by the app and mostly about the contents legibility by the users.
In chapter 5 ( Discussion and Conclusions, lines 424-432) the authors say that “During the phase of design, MoGeo App has been tested remotely and directly in the field on a group of people to get a first quick feedback on the usability of the App system and on the curiosity towards and liking of proposed contents. These tests have evidenced a broad appreciation of the App but also possible local difficulties related to the access of the telephone network. Regarding foreign tourists, this problem will probably be easily overcome by the possibility to change the mobile network operator in roaming conditions. For Italian tourists, instead, this solution may be not applicable, so a future implementation of the application is actually being evaluated to favour the off-line use of contents by allowing their download at least for areas with ascertained difficulty of internet access”.
They even recognize that the results would be more complete if tests were performed with tourists (lines 433-439): Further surveys on the reception and use of the App can certainly help improve it and are scheduled. In the next future we intend to conduct more systematic tests involving a group of families, to better consider the needs and interests of this specific target group and try to shorten the distance between contents for experts and lay people. This will be done by collecting through questionnaires the interest, appreciation and/or criticism expressed by the families, as well as the difficulties encountered in using the App and understanding of contents; in short, all the useful tips to improve the App.
Also because of that my opinion is that this work is incomplete and its final publication would benefit of results from a survey conducted with tourists, what would be the real test of the application.

Response 5: In section Discussion and Conclusions we have reported the major results of the expeditious tests we have carried out involving a dozen people without geological knowledge (lines 509-523), which we consider a first real, even not final test of the application. As already precised in the paper, we intend to do more targeted tests in the future with questionnaires to propose to a significant number of users.

Point 6: I consider also that the application could incorporate some innovative features that are being used and applied in the last years like Augmented Reality (AR) applied to geological crops and elements.

Response 6: We argued this within the section Discussion and conclusions, evidencing that it is a limitation of our application (lines 502-503). In the future we will try to add this innovative technique for some suitable sites.

Point 7: Besides, the app is limited to Android users, which eliminate a high quantity of tourists that use Apple devices, and it seems to be internet availability dependent at the sites.

Response 7: We have added the motivation of this choice in the discussion (lines 504-508).

Point 8: These issues (that I consider as weaknesses) should be at least incorporated in the discussion, recognizing the need to improve the tool. By the other hand, these are weaknesses that surely will be noticed by users in the field, what reinforce the importance of such component in the presented work.

Response 8: We have improved significantly the discussion, taking into account specifically the aforementioned suggestions.

Other minor remarks:

Point 9: Are the app contents only in Italian? Par example, in Fig. 6 (line 372-373) we can see the menu in English and the contents in Italian.

Response 9: The application benefits of a double language, as specified in the text (see lines 252-253). All contents (titles, texts in cards) have been created both in Italian and in English. However, we have improved and changed the figures 6 and 7 for that.

Point 10: Are the texts inside the figures with app print screens legible?

Response 10: Yes, they are. We suggest you visit the MoGeo Application, where it’s possible to note that. The password for the application is 1234.

Reviewer 3 Report

This is a highly-important paper because mobile applications can be really helpful to promote geotourism and to support geotourists travelling alone. The paper presents novel results, it is well-written and well-structured, and it is based on a representative set of the literature. I tend to recommend its acceptance after minor revisions. My recommendations are specified below.

  • Full affiliations of the authors should be provided (street address).
  • Key words: smarthphone -> smartphone
  • Add 3-5 citations to fresh general sources on geotourism to the first paragraph.
  • As aesthetics is mentioned (p. 8), some relevant works need to be considered.
  • Some of your geosites are viewpoint geosites – please, discuss how your application permits comprehension of distant features (see two basic references below).
  • In Discussion, you need to compare your tool with the others, earlier-proposed. At least, you need to give citations to the previous works where mobile applications are considered and to write about advantages of your application and/or matching the expectations of the potential users.
  • I'm not sure that all mandatory technical sections (between the main text and References) are provided in this manuscript.
  • The writing is perfect, but the language still needs some slight polishing.

References

General geotourism

Dowling, R.; Newsome, D. (Eds.) Handbook of Geotourism. Edward Elgar: Cheltenham, 2018.

Ólafsdóttir, R.; Tverijonaite, E. Geotourism: a systematic literature review. Geosciences 2018, 8, 234.

Ruban, D.A. Geotourism - A geographical review of the literature. Tourism Management Perspectives 2015, 15, 1-15.

Henriques, M.H.; Brilha, J. UNESCO Global Geoparks: a strategy towards global understanding and sustainability. Episodes 2017, 40, 349-355.

Justice, S.C. UNESCO global geoparks, geotourism and communication of the earth sciences: A case study in the Chablais UNESCO Global Geopark, France. Geosciences 2018, 8, 149.

Aesthetics

Kirillova, K.; Fu, X.; Lehto, X.; Cai, L. What makes a destination beautiful? Dimensions of tourist aesthetic judgment. Tourism Management 2014, 42, 282-293.

Mikhailenko, A.V.; Nazarenko, O.V.; Ruban, D.A.; Zayats, P.P. Aesthetics-based classification of geological structures in outcrops for geotourism purposes: a tentative proposal. Geologos 2017, 23, 45-52.

Viewpoint geosites

Migoń, P.; Pijet-Migoń, E. Viewpoint geosites – values, conservation and management issues. Proceedings of the Geologists' Association 2017, 128, 511–522.

Mikhailenko, A.V.; Ruban, D.A. Environment of Viewpoint Geosites: Evidence from the Western Caucasus. Land 2019, 8, 93.

Author Response

Authors: We thank the reviewer for his valuable suggestions. All the manuscript corrections are reported below, describing them point by point. To install the MoGeo App the password is: 1234.

Point 1: This is a highly-important paper because mobile applications can be really helpful to promote geotourism and to support geotourists travelling alone. The paper presents novel results, it is well-written and well-structured, and it is based on a representative set of the literature. I tend to recommend its acceptance after minor revisions.
My recommendations are specified below.

Response 1: We thank the reviewer for all his suggestions.

Point 2: Full affiliations of the authors should be provided (street address).

Response 2: Street address was added (line 5).

Point 3: Key words: smarthphone -> smartphone

Response 3: Done (line 26).

Point 4: Add 3-5 citations to fresh general sources on geotourism to the first paragraph.

Response 4:  We have added some relevant citations to the first part of the introduction (lines 30-36).

Point 5: As aesthetics is mentioned (p. 8), some relevant works need to be considered.

Response 5: We have added a specific consideration on this point in section Materials and methods, considering relevant works (see citations in the text) (lines 222-234).

Point 6: Some of your geosites are viewpoint geosites – please, discuss how your application permits comprehension of distant features (see two basic references below).

Response 6: We have added a discussion on these issues in section Materials and methods, considering relevant works (see citations in the text) (lines 277-298).

Point 7: In Discussion, you need to compare your tool with the others, earlier-proposed. At least, you need to give citations to the previous works where mobile applications are considered and to write about advantages of your application and/or matching the expectations of the potential users.

Response 7: We have added a part of text that responds to this request (lines 488-503).

Point 8: I'm not sure that all mandatory technical sections (between the main text and References) are provided in this manuscript.

Response 8: The missing mandatory technical sections has been added (lines 546-549).

Point 9: The writing is perfect, but the language still needs some slight polishing.

Response 9: We have accurately revised the text in terms of language and writing style.

Round 2

Reviewer 2 Report

Dear authors,

I am pleased to see that the main remarks made by the reviewers were accepted. Therefore, I think that the manuscript was significantly increased.